# Static Culture Combined with Aeration in Biosynthesis of Bacterial Cellulose

**DOI:** 10.3390/polym13234241

**Published:** 2021-12-03

**Authors:** Nadezhda A. Shavyrkina, Ekaterina A. Skiba, Anastasia E. Kazantseva, Evgenia K. Gladysheva, Vera V. Budaeva, Nikolay V. Bychin, Yulia A. Gismatulina, Ekaterina I. Kashcheyeva, Galina F. Mironova, Anna A. Korchagina, Igor N. Pavlov, Gennady V. Sakovich

**Affiliations:** 1Bioconversion Laboratory, Institute for Problems of Chemical and Energetic Technologies, Siberian Branch of the Russian Academy of Sciences (IPCET SB RAS), 659322 Biysk, Russia; 32nadina@mail.ru (N.A.S.); eas08988@mail.ru (E.A.S.); sitnikova97.97@mail.ru (A.E.K.); evg-gladysheva@yandex.ru (E.K.G.); nbych@yandex.ru (N.V.B.); julia.gismatulina@mail.ru (Y.A.G.); makarova@ipcet.ru (E.I.K.); yur_galina@mail.ru (G.F.M.); yakusheva89_21.ru@mail.ru (A.A.K.); pawlow-in@mail.ru (I.N.P.); admin@ipcet.ru (G.V.S.); 2Biysk Technological Institute, Polzunov Altai State Technical University, 659305 Biysk, Russia

**Keywords:** static culture, aeration, bacterial cellulose, *Medusomyces gisevii* Sa-12

## Abstract

One of the ways to enhance the yield of bacterial cellulose (BC) is by using dynamic aeration and different-type bioreactors because the microbial producers are strict aerobes. But in this case, the BC quality tends to worsen. Here we have combined static culture with aeration in the biosynthesis of BC by symbiotic *Medusomyces gisevii* Sa-12 for the first time. A new aeration method by feeding the air onto the growth medium surface is proposed herein. The culture was performed in a Binder-400 climate chamber. The study found that the air feed at a rate of 6.3 L/min allows a 25% increase in the BC yield. Moreover, this aeration mode resulted in BC samples of stable quality. The thermogravimetric and X-ray structural characteristics were retained: the crystallinity index in reflection and transmission geometries were 89% and 92%, respectively, and the allomorph Iα content was 94%. Slight decreases in the degree of polymerization (by 12.0% compared to the control―no aeration) and elastic modulus (by 12.6%) are not critical. Thus, the simple aeration by feeding the air onto the culture medium surface has turned out to be an excellent alternative to dynamic aeration. Usually, when the cultivation conditions, including the aeration ones, are changed, characteristics of the resultant BC are altered either, due to the sensitivity of individual microbial strains. In our case, the stable parameters of BC samples under variable aeration conditions are explained by the concomitant factors: the new efficient aeration method and the highly adaptive microbial producer―symbiotic *Medusomyces gisevii* Sa-12.

## 1. Introduction

Bacterial cellulose (BC) is a polymer that is synthesized by microorganisms. Glucose is the monomer of BC, as for cellulose of plant origin. BC is distinct in the 3D architecture of the network structure of ultrafine fibers whose thickness varies from 20 to 100 nm [1,2,3,4,5,6]. BC contains constitutionally no lignin and hemicellulose impurities [7] and exhibits high values of physicochemical characteristics such as mechanical strength, water-holding capacity and crystallinity. Moreover, the fundamental property of BC is its biocompatibility [1,2,3,4,5,6]. BC is used in various fields ranging from traditional technical and food industries to biomedicine. The use of BC as a part of the newest composites using the emerging technologies [8] is continuously expanding the application range of this magnificent polymer [2,6,7,9,10].

The BC-producing bacteria are strict aerobes; therefore, the amount of oxygen dissolved in the growth medium has an impact on the biosynthesis of BC. Under static culture conditions, oxygen is a limiting factor for the ongoing intracellular metabolic processes and BC biosynthesis, and influences physicochemical properties of BC [11,12]. The oxygen deficiency of the culture medium causes the microbial cell growth and BC biosynthesis to cease.

To overcome the said problem, researchers are being proposed to use dynamic culture and adopt different-type bioreactors for the scaled manufacture of BC.

Dynamic culture takes place with a forced oxygen feed into the growth medium stratum, resulting in BC particles of different sizes (10 µm to 10 mm wide) and of different shapes (sphere-like, ellipsoid-like, star-like, fibrous suspensions, pellets or irregular masses). That is, the growth medium undergoes perturbations under the dynamic culture, impeding the formation of an a BC gel-film in its in its entirety [1]. Exclusive applications were proposed for sphere-like BC samples, for example, for improved physical absorption, transport, and cross-linking of different compounds [13,14]. In addition, BC obtained by dynamic culture can be used, if there are no BC size and shape requirements, for example, in the manufacture of paper, desserts, in the design of composites, including 3D printing. However, dynamic culture generally leads to limited application fields [1].

It was shown in the studies [15,16] that an excessive oxygen feed reduces the yield of BC. Thus, despite the increased rate of oxygen delivery into the nutrient medium, the dynamic and static culture methods at the same duration provide equal yields of BC. The researchers ascribe the low BC yield to the resulting non-cellulosic mutants and genetically instable bacteria under the dynamic conditions [17,18]. Moreover, an alteration of physicochemical properties of BC, specifically the microstructure of BC fibers, a decrease in the polymerization degree and crystallinity index, and an impairment of mechanical behavior were observed [19,20].

The use of different-type bioreactors made it possible to improve the BC yield [21,22]; however, as in the case of dynamic culture, the appearance of BC changed and its characteristics deteriorated [23,24]. So, it is generally accepted that the use of static culture is preferable to preserve the physicochemical properties of BC [1,25].

This is the first study to have combined the static biosynthesis of BC with forced aeration. A new aeration method by feeding the air onto the growth medium surface rather than deep into the medium is proposed herein. We hypothesized that this would allow the BC quality to be stable, which is warranted only by static culture, and the BC yield to be enhanced by supplemental aeration.

## 2. Materials and Methods

### 2.1. Microbial Producer

*Medusomyces gisevii* Sa-12 (Scientific Center “Kurchatov Institute”–Research Institute for Genetics and Selection of Industrial Microorganisms, Moscow, Russia) was chosen as the primary culture for the biosynthesis of BC. This symbiotic culture, commonly known as *Kombucha* or tea fungus, is comprised of 10 genera of acetobacteria, which, in fact, produce a cellulosic gel-film, and about 25 genera of yeasts that provide a comfortable co-existence of BC-producing microorganisms. The composition of the culture has repeatedly been described in the literature [26,27].

### 2.2. Biosynthesis of Bacterial Cellulose

For the biosynthesis of BC, we used a semisynthetic nutrient medium that was prepared as follows: ionized water was brought to boil, and dry bohea black tea (at 5 g/L, which is equivalent to a black tea extractive content of 1.6 g/L) was added, infused for 15 min and filtered off, and then 20 g/L glucose was added (OOO Promsintez, CAS No. 5996-10-1, Chapayevsk City, Russia). Tea is a mandatory ingredient of nutrient media for the symbiotic culture [28]. The optimum concentrations of glucose and tea to warrant a high yield of BC and a low formation of byproducts were identified in our previous study [29].

A culture medium obtained after 14-day cultivation of symbiotic *Medusomyces gisevii* Sa-12 on the said nutrient medium was employed as the inoculum. The inoculum volume was 10% of the nutrient medium volume; the yeast cell count was 13.0 × 10^6^ in 1 mL, and the acetobacteria cell count was 2.1 × 10^6^ in 1 mL.

Biosynthesis of BC was performed under static conditions but with forced air feeding. The culture was run in a Binder-400 climate chamber (Berlin, Germany) which has a thermostat function and is fitted with a main air supply line to the bottom of the chamber. The circulation and uniform distribution of the air was due to a fan located on the top of the chamber, and due to perforated side walls. The constant temperature conditions were thus maintained within the chamber space. The air feed rates in the chamber were 3.3 L/min, 6.3 L/min, 9.2 L/min, 12.3 L/min and 16.7 L/min. These were equivalent to the air change rates of 0.5, 0.9, 1.4, 1.8 and 2.5 of the chamber volumes per hour, respectively. The climate chamber equipped with high efficiency particulate air filters was installed in a clean room. As the control, biosynthesis was run without aeration. The culture temperature was 27 °C, pH was not controlled, and the culture time was 24 days.

The culture was performed in 250 mL plastic containers. For each of the aeration modes, the inoculum was injected into the nutrient medium volume (16 L), afterwards a 200 mL culture medium was injected into 80 containers each. Ten containers were placed onto 8 chamber shelves each―representing widely spaced grids little impeding the air stream. Onto the chamber’s bottom were put 8 ionized water containers (200 mL water volume) to create the desired air humidity and reduce evaporation losses. Samples were collected every business day; the entire container was sampled. An ionized water container was put into the place of the growth medium container each that was removed. The air humidity was measured by a TK-5 contact thermometer (OOO Techno-AS, Kolomna city, Russia). The air humidity was 89−94% throughout the experiment (24 days).

After the biosynthesis was complete, BC gel-films were removed from the growth medium and washed with water until the microbial cells and coloring agents of the nutrient medium were removed in full, by the procedure described [29].

The experimental data for each of the aeration rates were obtained in triplicate; the results were expressed as the mean and experimental errors (standard deviation in sampling).

### 2.3. BC Yield

The BC yield was calculated by Equation (1), where the BC yield was estimated as the quantity of the resultant dry BC with respect to the initial concentration of the carbon source–glucose:(1)$$\eta =\frac{m_{\mathrm{BNC}}}{C_{g}\cdot V\cdot 0.9}\cdot 100$$
where *η* is the BC yield, %; *m*_BNC_ is the BC sample weight on an oven-dry basis (g), *C*_g_ is the glucose concentration in the medium (g/L), *V* is the medium volume (L), and 0.9 is the conversion coefficient attributed to the water molecule detachment upon polymerization of glucose into cellulose. The yield was calculated by the procedure reported [30].

### 2.4. Analytical Techniques

#### 2.4.1. Active Acidity

The active acidity during the biosynthesis of BC was measured by an I-160 MI ion meter (OOO Izmeritelnaya Tekhnika, Moscow, Russia).

#### 2.4.2. Concentration of Reducing Sugars

The level of reducing sugars (calculated in glucose equivalent) in the growth medium was controlled using 3,5-dinitrosalicylic acid (Panreac, CAS No. 609-99-4, Barcelona, Spain), which changed its color from yellow to red when reacted with the reducing sugar (RS). The coloring intensity and, correspondingly, the RS concentration were determined on a UNICO UV-2804 spectrophotometer (United Products & Instruments, Dayton, NJ, USA) at a wavelength of 530 nm.

#### 2.4.3. Scanning Electron Microscopy of BC

The degree of polymerization of the BC samples was measured by a viscometer using cadoxene as solvent (ethylenediamine, AO LenReaktiv, CAS No. 107-15-3, Saint-Petersburg, Russia; cadmium oxide, AO LenReaktiv, CAS No. 1306-19-0, Saint-Petersburg, Russia), as described [31].

#### 2.4.4. BC Degree of Polymerization

Scanning electron microscopy (SEM) images to examine the microfibrillar structure of BC were acquired on a JSM-840 scanning electron microscope (JEOL Ltd., Tokyo, Japan).

#### 2.4.5. Thermomechanical Analysis

The strength behavior of BC was measured on a Shimadzu DTG-60 thermomechanical analyzer (Kyoto, Japan): the test specimen was stretched at a rate of 5.0 g/min starting from 0.0 g to the maximum load of 400.0 g until failure, at a temperature of 23.0 °C.

The thickness of the BC samples was measured by a ICh-10 I-class dial indicator thickness gauge (Kirov Factory “Kirovskiy Instrumentalshchik”, Kirov city, Russia). A Shimadzu TMA-60 instrument (Kyoto, Japan) was used to measure the strength of the BC samples. The test samples were stretched at a rate of 5 g/min to the maximum load of 500 g, at room temperature.

The Young’s modulus of elasticity was estimated by Equation (2):(2)E=σ_sx/((ε_sx/100))
where *E* is the Young’s modulus of elasticity, MPa; σ_sx is the conventional yield limit, MPa; ε_sx is the relative elongation at yield, %.

#### 2.4.6. Thermogravimetric Analysis

Thermogravimetric analysis was performed on a Shimadzu DTG-60 thermomechanical analyzer (Kyoto, Japan) under the following conditions: the test specimen was heated at a rate of 10 °C/min to the maximum temperature of 600 °C in nitrogen environment at a gas flowrate of 40 mL/min.

#### 2.4.7. X-ray Diffraction of BC

X-ray structural characterization was performed on a DRON-6 monochromatic diffractometer (NPO Burevestnik, Moscow, Russia) with Fe*K*_α_ radiation at scattering angles of 3 to 145°. The spectral characteristics were calculated with the PdWin software package. The index of crystallinity and Iα allomorph content were determined by the procedures reported [32,33].

The analyses were carried out using equipment provided by the Biysk Regional Center for Shared Use of Scientific Equipment of the SB RAS (IPCET SB RAS, Biysk, Russia).

## 3. Results and Discussion

### 3.1. Variation of Culture Parameters during BC Biosynthesis

#### 3.1.1. Active Acidity and RS Concentration

Figure 1 displays the variations in active acidity level and reducing sugar (RS) concentration when *Medusomyces gisevii* Sa-12 was cultured at different aeration modes.

A decrease in the active acidity of the culture medium during the cultivation is typical of the *Medusomyces gisevii* Sa-12 microbial producer, and is explained by acetic, succinic, malic and gluconic acids being formed as metabolic products of the acetobacteria and yeasts [34]. The experimental data obtained herein illustrate a similar pattern of the decrease in pH of the culture medium, irrespective of the aeration rate. It should be noted in this case that the acidity of the samples declined most intensively without forced aeration and at a minimum aeration rate of 3.3 L/min at the onset of the cultivation in the initial two days.

During the RS consumption, the first stage of fast RS utilization (from 0 to 10 days of cultivation) and the second stage of slow utilization (after 10 days of cultivation) were observed, which is typical of the *Medusomyces gisevii* Sa-12 microbial producer [29].

#### 3.1.2. BC Yield

Figure 2 illustrates the BC yield as a function of the aeration mode.

The maximum BC yield of 9.1% was obtained at an air feed rate of 6.3 L/min. That is, this allowed a 25% increase in the BC yield as compared to the control. A decrease or further increase in the air feed rate caused the BC yield to decline to the baseline (7.2%). When the aeration rate was lowered below 6.3 g/L, the BC yield reduction is explained by aerial oxygen deficiency that limited the BC biosynthesis process. When the aeration rate was raised above 6.3 L/min, the decreased BC yield is explained by the negative impact of excessive oxygen concentrations, which is in a good agreement with the literature data [15,16].

Kouda et al. [20] reported a 12.8% decrement in the BC yield when the partial oxygen pressure rose from 0.25 to 0.55 atm due to an increase in the air stream from 0.25 to 1.0 vvm.

Krusong et al. in their recent study [35] proposed a quite sophisticated way to overcome the known problem occurring during the aeration. The problem is that an increase in oxygen concentrations adversely affects the BC yield. The study suggested that a luffa sponge be used as a matrix and carboxymethylcellulose be added to the medium in order to mitigate the negative impact of excessive oxygen concentrations. The combination of these strategies allowed the authors to attain a BC yield comparable to that obtained under static culture.

In our study, the new aeration method by feeding the air onto the growth medium surface made it possible to achieve a progress in the BC technology. This simple strategy appeared to be a perfect alternative to the dynamic aeration method. Besides, the static culture combined with aeration furnished a higher BC yield than static culture alone.

### 3.2. Characterization of BC Samples

#### 3.2.1. Appearance of BC Gel-Films

The fed air velocity differed considerably on different chamber shelves with forced aeration by which the air was fed on the growth medium surface. In the absence of aeration, as well as in climatic chamber sections located far away from the air streams, uniform BC pellciles were formed (Appendix A). In those chamber sections where air circulation was intensive (on the top of the chamber, near the fan), inhomogeneous BC gel-films were observed (Appendix A), as if a wave-like motion of the liquid was imprinted on the culture. Thus, even though the air was fed onto the surface rather than deep into the medium, distortions of the homogeneous structure of BC gel-films could not be avoided.

Table 1 summarizes a proportion of uniform gel-films depending on the aeration mode: the higher the air feed rate, the lower the proportion. When the air feed rate was raised above 6.3 L/min, the proportion of uniform gel-films became less than 70%―which is undesirable.

#### 3.2.2. Microstructure of BC

The ultrafine reticulate structure of BC nanofibers is an important fundamental property of BC [36]. Figure 3 shows SEM images of the reticulate structure of BC samples obtained with different air feed rates.

Figure 3 illustrates that thickening or adhesion of BC nanofibers occurs at different aeration modes, with no clear relationship being traceable between the BC structure alteration and the aeration rate. However, the 3D irregular reticulate structure typical of BC is retained principally at all rates of air feeding onto the culture medium surface. This is a positive result.

The average widths of BC microfibrils (estimated with Image J software) are listed in Table 2. The relationship between the microfibril width and aeration rate was not found. A slight decline in the width of microfibrils was observed under aeration, but given the root-mean-square deviation value, this decline cannot be considered significant.

In contrast to the new aeration method proposed herein, the microstructure of BC is altered irreversibly when the air is fed deep into the culture medium [37]. The location of fibrils in agitated cultures is more widely spaced with a larger pore size. The formation of microfibrils is probably impaired by agitation during the fermentation process.

#### 3.2.3. BC Degree of Polymerization

The highest degree of polymerization at 4300 was documented with no forced aeration. The degree of polymerization declined as the rate of air feeding onto the culture medium surface was raised. At the chosen aeration rate of 6.3 L/min, the degree of polymerization was 3800, which is 12% lower that than in the control. When the air feed rate was elevated above 6.3 L/min, the degree of polymerization diminished dramatically. At the maximum air feed rate of 16.7 L/min, the degree of polymerization was the lowest, 950 or only 22% of the control value (Table 2).

This can be explained by activation of oxidative processes in cells of BC microbial producers and by disturbance of the function of enzyme systems. A decline in the degree of polymerization was noticed in other studies [19,23,24] when dynamic culture and bioreactors were used.

#### 3.2.4. BC Elastic Modulus

Similarly to the changes in the degree of polymerization, the elastic modulus was observed to decline as the aeration rate was elevated. The elastic modulus was 910 MPa without aeration, while it decreased 2.9-fold and was 315 MPa at a maximum aeration rate of 16.7 L/min. Also, the critical point, after which the elastic modulus was observed to decline sharply, was when the air feed rate was raised above 6.3 L/min (Table 2). At an aeration rate of 6.3 L/min, the elastic modulus was 795 MPa, which is 12.6% lower than that of the control.

#### 3.2.5. Thermogravimetric Analysis (TGA)

TGA data dependences on the aeration mode were not observed (Table 2). The highest onset temperature of decomposition was for the sample obtained at an air flow rate of 6.3 L/min, that is, the sample was the most thermally stable and purest [38]. The other values of BC decomposition temperature and weight loss did not correlate between each other and did not depend on the aeration mode. But it can be said in summary that these are close values evidencing more of similarities of the samples rather than their dissimilarities.

It is generally believed that the behavior of the sample when thermally decomposed is influenced by factors such as molecular weight, crystallinity, degree of polymerization, and compactness of structural interlacement [39].

We believe that the impurities, but not the BC structure, affect most notably the TGA analysis. Because the proportion of inhomogeneous BC gel-films increased as the air feed rate was raised, it is the structural non-uniformity that governs the unequal quality of BC washing free of impurities.

The comparison of the TGA data for the BC samples obtained in this study with those reported in the literature demonstrated that the results are alike [38].

#### 3.2.6. X-ray Diffraction

The index of crystallinity varied within the range from 87% to 93% in reflection geometry and from 89% to 95% in transmission geometry, and the allomorph Iα content ranged from 94% to 98%, regardless of the aeration mode. With the chosen aeration mode of 6.3 L/min, the indices of crystallinity were 89% in reflection geometry and 92% in transmission geometry, and the allomorph Iα content was 94%. The size of crystallites did not depend on aeration rate (Table 2), as in the study [40] in which the size of crystallites did not change when the biofilm reactor was used. It can be inferred that the X-ray structural characteristics of BC samples do not depend on the aeration mode.

The literature describes a considerable decrease in the BC index of crystallinity when dynamic aeration methods are used. It is indicated that the structural variation of BC is accompanied by the alteration of its physicochemical properties [17,20,24].

The fact that the basic X-ray structural characteristics of BC samples are independent of the aeration mode, as established in the present study, is explained not only by the optimum aeration conditions found, but also by the properties of the microbial producer used―*Medusomyces gisevii* Sa-12. Our previous study discovered that the quality of BC produced by *Medusomyces gisevii* Sa-12 under unsterile conditions does not depend on the composition of hydrolyzate media. The hydrolyzate media were derived from *Miscanthus* by four chemical pretreatment methods, followed by enzymatic hydrolysis [41].

Orlovska et al. [42] investigated the behavior of the microecosystem of the tea fungus culture (i.e., *Medusomyces gisevii* Sa-12) and the quality of the produced BC, through the means of simulating the Mars-like environment. They documented a high survivability of the tea fungus culture and a decrease in the BC yield, but the structural characteristics of BC in this case showed no significant changes in all of the trials.

Thus, the *Medusomyces gisevii* Sa-12 symbiotic culture is capable of synthesizing stable-quality BC, irrespective of the cultivation conditions such as the composition of media used, sterility-free media, aeration modes, and biosynthesis under extreme conditions. By the totality of factors, symbiotic *Medusomyces gisevii* Sa-12 demonstrates undisputable merits for industrial application.

Forte et al. [43] note that the manufacture of BC by using individual strains under production environments is quite limited, whereas the production of BC by using symbiotic cultures is traditionally widely practiced in the Asian-Pacific Region. The production of BC by a using symbiotic culture currently continues to broaden, including that for technical applications [25,44,45,46].

The BC gel-films produced in the present study were employed to fabricate composite paper using hardwood pulp [47]. We noted that despite the resultant BC gel-films being inhomogeneous, their basic properties such as moisture, water-holding capacity, and X-ray structural properties (allomorph Iα content and index of crystallinity) remained stable. Because the samples were ground for use as a part of the composites, the differences in parameters such as degree of polymerization and elastic modulus are not critical.

## 4. Conclusions

Here we have successfully combined static biosynthesis of BC with forced aeration. The new aeration method by which the air is fed onto the culture medium surface at a rate of 6.3 L/min allows a 25% increase in the BC yield. The biosynthesis performed at this aeration rate furnished BC samples of stable quality. The thermogravimetric and X-ray structural characteristics remained unchanged: the index of crystallinity was 89% in reflection geometry and 92% in transmission geometry, and the allomorph Iα content was 94%. The decline in the degree of polymerization by 12.0% (3800 versus 4300) and in elastic modulus by 12.6% (795 MPa versus 910 MPa), when compared to the control with no aeration, is not critical. The stability of the parameters of BC samples under varying aeration conditions is explained by the concomitant factors: the new effective aeration method and the high adaptivity of the microbial producer―symbiotic *Medusomyces gisevii* Sa-12.

## Figures and Tables

**Figure 1 polymers-13-04241-f001:**
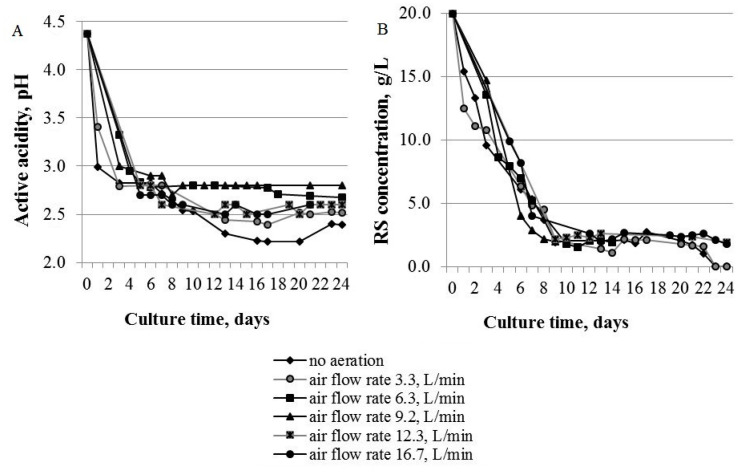
(**A**) Active acidity level and (**B**) reducing sugar (RS) concentration plotted against different aeration rates. The half-width of the confidence interval for RS concentration was ±0.2 g/L, pH ± 0.1.

**Figure 2 polymers-13-04241-f002:**
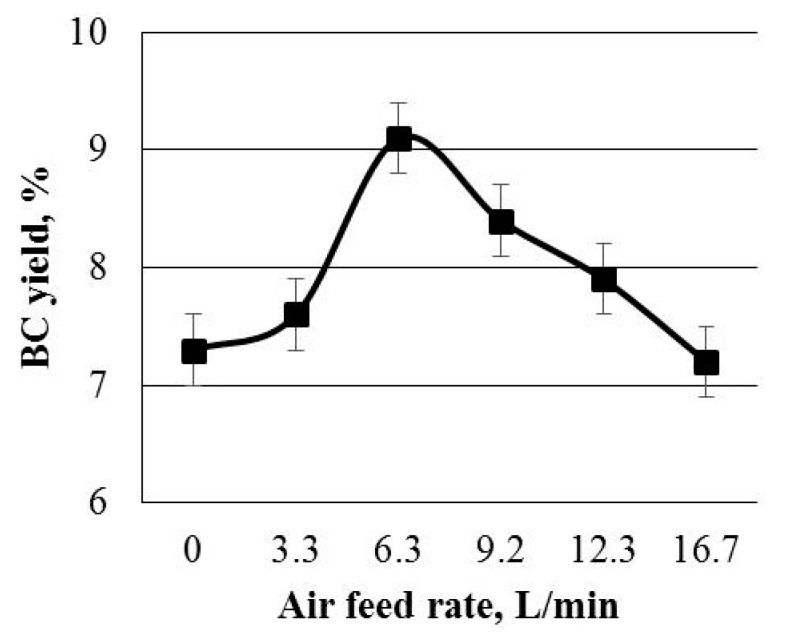
The bacterial cellulose (BC) yield plotted against the aeration rate.

**Figure 3 polymers-13-04241-f003:**
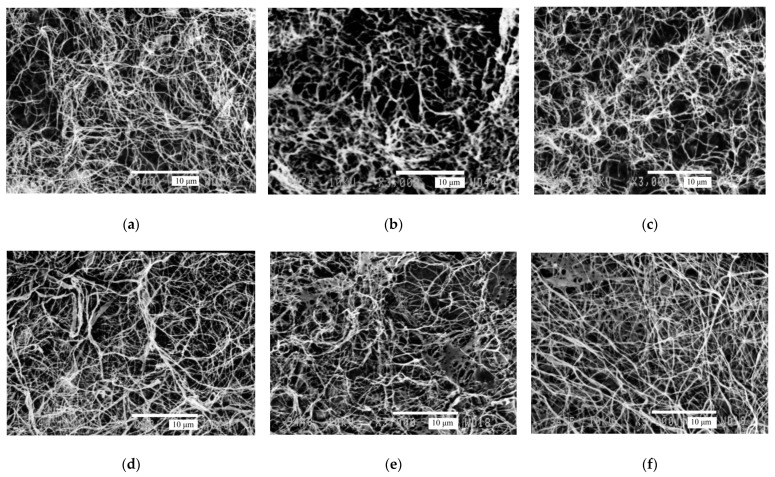
Scanning electron microscopy (SEM) micrograms of BC samples obtained at different air flow rates (L/min) (zoom ×3000) (**a**) 0 (no aeration), (**b**) 3.3 L/min, (**c**) 6.3 L/min, (**d**) 9.2 L/min, (**e**) 12.3 L/min, (**f**) 16.7 L/min.

**Table 1 polymers-13-04241-t001:** Bacterial cellulose (BC) yield and proportion of uniform gel-films subject to aeration mode.

Aeration Rate, L/min	Proportion of Uniform Gel-Films, %	BC Yield, %
No aeration (control)	100	7.3 ± 0.3
3.3	89.9 ± 1.5	7.6 ± 0.3
6.3	72.4 ± 2.2	9.1 ± 0.3
9.2	64.9 ± 1.9	8.4 ± 0.3
12.3	60.7 ± 2.5	7.9 ± 0.3
16.7	53.2 ± 2.6	7.2 ± 0.3

The structural non-uniformity of BC gel-films limits their application fields.

**Table 2 polymers-13-04241-t002:** Basic physicochemical characteristics of BC samples obtained at different aeration rates.

Parameter	Aeration Rate, L/min					
	No Aeration	3.3	6.3	9.2	12.3	16.7
Average microfibril width, ±40 nm	107	93	81	101	85	97
Degree of polymerization, ±100	4300	4200	3800	2550	2100	950
Young’s modulus, ±10 MPa	910	860	795	490	360	315
TGA summary data						
Sample weight loss at the first stage, ±0.5%	4.1	5.0	3.9	4.5	4.1	2.8
Sample weight loss at the second stage (within sample decomposition range), ±0.1%	7.8	17.9	12.4	19.5	37.4	19.1
Onset temperature of decomposition, ±5 °C	353	372	374	330	308	297
Sample weight loss at the third stage, ±1.0%	62.5	49.2	63.6	52.4	30.4	45.7
Sample weight loss at the fourth stage, ±0.5%	8.4	9.1	5.9	6.0	9.0	18.3
Unburnable residue, ±1.0%	17.2	18.9	14.3	17.6	19.1	14.2
Concentrations of cellulose allomorphs and crystallinity index as measured by X-ray diffraction						
Iα-allomorph, ±5%	98	94	94	96	94	97
Iβ-allomorph, ±5%	2	6	6	4	6	3
Index of crystallinity (reflection geometry), ±5%	93	93	89	87-90	93	91
Index of crystallinity (transmission geometry), ±5%	89	95	92	92	93	92
Crystallite size of <110>, ±0.5 nm	7.7	7.8	7.8	7.8	7.8	7.7

## Data Availability

The data presented in this study are available upon request from the corresponding author.

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
