# Peer review of "Static Culture Combined with Aeration in Biosynthesis of Bacterial Cellulose"

_polymers, 2021, doi:10.3390/polym13234241_

Round 1

Reviewer 1 Report

Although the work is well written, discussions are superficial. In addition, there are no impacting novelties in the proposal and results of this work. For that reason, I do not recommend the article for publication.
The large number of authors involved (12) in a relatively simple work with little impact is noteworthy.

The standard error in the results of Active Acidity and RS concentration was not informed.

Author Response

The response has been uploaded as a PDF file.

Reviewer 2 Report

Dear authors article entitled Static Culture Combined with Aeration in Biosynthesis of Bacterial Cellulose is interesting. The manuscript is well written and describe a simple  but effective solution for improvement of BC production process effectiveness. However before publication in Polymers Journal some elements of manuscript could be improved.

Comments;

1. The analytical techniques could be listed with clear subsections.

2. The legend of figure 1b should be moved to graph window.

3. Authors performed SEM analysis and according the pictures they could perform analysis of  dimensions of microfibrils by using for example Image J software. This results could improve significance of this analysis that is moderate in presented form.

4. The main carbon source in fermentation media was a glucose. The main problem with this substrate is its conversion to gluconic acid by glucose oxidase and fast acidification of cultivation media. The acetic acid and malic acid especially in initial time of fermentation do not contribute significantly in acidification of cultivation media. The high aeration can increase speed of this process (more substrate – oxygen - for glucose oxidase) and decrease yield of process. In this case another substrates could be more suitable for example fructose. Could authors discus this problem more widely.

5. The crystallites size should be added to results in table 2.

6. Why authors did not check more parameters of BC such as water holding capacity, swelling ratio, retention ability ? This parameters are especially important for BC that has a high potential to be used in biomedical applications.

Author Response

(The authors gave the same response as above.)

Reviewer 3 Report

The work is written with great care technically and substantively. It is one of the few works where the authors clearly indicate the origin of kombucha. The innovative aspect is the detail of the bioreactor access to oxygen. Bacterial cellulose has a real chance of replacing dangerous fossil-based plastics in the future which is why such work should be published. I believe that this work should be published after taking into account the minor comments below.  

Can you add measured value error to values in table 2.

The term "pellicles" used by the authors is not necessarily appropriate here. It is used in gastronomy and often refers to the protein layer, so you might consider using a different term here.

Author Response

(The authors gave the same response as above.)

Round 2

Reviewer 1 Report

No comments.